# Strategies and Patterns of Codon Bias in Molluscum Contagiosum Virus

**DOI:** 10.3390/pathogens10121649

**Published:** 2021-12-20

**Authors:** Rahul Raveendran Nair, Manikandan Mohan, Gudepalya R. Rudramurthy, Reethu Vivekanandam, Panayampalli S. Satheshkumar

**Affiliations:** 1Centre for Evolutionary Ecology, Aushmath Biosciences, Vadavalli Post, Coimbatore 641041, India; 2College of Pharmacy, University of Georgia, Athens, GA 30605, USA; Manikandan.Mohan@uga.edu; 3Poxvirus and Rabies Branch, Centers for Disease Control and Prevention, Atlanta, GA 30329, USA; murthygr@fndr.in; 4Department of Biotechnology, Bharathiyar University, Coimbatore 641046, India; reethu.res@gmail.com

**Keywords:** molluscum contagiosum virus (MCV), synonymous codon usage bias (SCUB), selection, mutation

## Abstract

Trends associated with codon usage in molluscum contagiosum virus (MCV) and factors governing the evolution of codon usage have not been investigated so far. In this study, attempts were made to decipher the codon usage trends and discover the major evolutionary forces that influence the patterns of codon usage in MCV with special reference to sub-types 1 and 2, MCV-1 and MCV-2, respectively. Three hypotheses were tested: (1) codon usage patterns of MCV-1 and MCV-2 are identical; (2) SCUB (synonymous codon usage bias) patterns of MCV-1 and MCV-2 slightly deviate from that of human host to avoid affecting the fitness of host; and (3) translational selection predominantly shapes the SCUB of MCV-1 and MCV-2. Various codon usage indices viz. relative codon usage value, effective number of codons and codon adaptation index were calculated to infer the nature of codon usage. Correspondence analysis and correlation analysis were performed to assess the relative contribution of silent base contents and significance of codon usage indices in defining bias in codon usage. Among the tested hypotheses, only the second and third hypotheses were accepted.

## 1. Introduction

In universal genetic code, any given amino acid except tryptophan and methionine is encoded by a specific set of multi-fold degenerate codons called synonymous codons [1,2]. As an event of mutation which causes replacement of one synonymous codon with another in a given coding region does not modify the amino acid sequence, these mutations are called ‘silent’ [3]. Although these synonymous changes are seemingly neutral, selection of synonymous codons occurs during the process of evolution as these ‘silent’ changes have many effects on the functioning of a living cell [3]. Due to selection, even though translational mechanisms in organisms are relatively conserved from pole to pole, patterns of synonymous codon usage (SCU) are non-random across species, resulting in species-specific SCU [4,5]. Further, usage of synonymous codons varies within genes of the same genome [6,7,8]. 

Despite the fact that selection and mutation still remain as two major explanations in delineating the origin of SCUB (SCU bias) [5,9], several factors of varying intensities contribute to the origin of distinct patterns of SCU within and between genomes [10], for instance, GC content [11,12], rate of gene expression [13,14], mRNA decoding tempo of ribosomes [13], mRNA secondary structure [15,16], mRNA turnover [17,18], co-translational protein folding and translation elongation [19], gene function [20], rate of recombination [21,22], gene length [23,24], codon position [21], habitat stress [25,26] and population size [21].

Intraspecies SCUB is often viewed as the result of selection because the higher the number of preferred codons, the higher the level of gene expression would be [23,27]. In contrast, mutational pressure is assumed to be the primary player in determining interspecies SCUB [1,28,29]. However, such generalizations of driving forces behind SCUB in intraspecies and interspecies scenarios are not yet fully justified [30,31] as compositional constraints (differential nucleotide contents) of genomes are also crucial. For instance, GC-rich genomes tend to favor G and C ending codons whereas AT-rich genomes preferentially use A and T ending codons [6,32,33]. Research on SCUB in various species unveiled the role of weak selection acting at the molecular level towards molecular evolution [34,35,36], and such studies produced substantial evidence to develop molecular evolutionary models based on selection other than neutral molecular evolution model [37,38]. An understanding of differential influences of these forces on shaping SCUB in a species is of paramount importance to research as it paves way for studying the evolutionary potential of genomic machinery of that species.

Viruses are parasites which depend on host cells to undertake key biomolecular measures of survival, such as transcription, translation and replication [39]. Viral genes are capable of altering various steps in the pathogen identification pathways of host cell [40]. Certain viruses are proposed to remain in host cell for long durations without being identified by host immune mechanisms and may follow a relaxed inexorable way of reproduction using cell’s replication machinery [39]. Essentially, such long-term association in host cells can cause transformation of whole viral genome (DNA/RNA) as an integral part of the host genome (colonization), which will decide the direction of the evolution of the host. Analyses of SCUB of various viral genomes reported that the efficiency of adaptation of viral genomes to the host is directly proportional to rate of similarity of SCUB between virus and host; the more the similarity, the higher the adaptation will be [41,42]. A recent study revealed that optimum SCUB pattern of viral genome follows slight deviation from the SCUB pattern of the natural host in order to avoid excessive expression and depletion of the tRNA pool as host fitness is important for the virus to survive in the natural host/virus systems [43]. Although debatable, the concept that viruses develop unique genes and then colonize bacterial and vertebrate lineages reveals the evolutionary significance of viruses [39]. Hence, studying SCUB patterns of viral genomes will help to gain significant insights into overall viral sustenance, codon adaptability and viral pathogenesis with respect to natural and symptomatic hosts [44]. 

Molluscum contagiosum virus (MCV) is a double-stranded DNA virus belonging to the genus Molluscipox of *Poxviridae* family [45]. Molluscum contagiosum (MC) is a self-limited skin disease caused by MCV in humans which is characterized by small but raised mollusca (lesions) on the top layer of skin [46]. High incidence of MC is limited to the pediatric population, but immunodeficient individuals and sexually active adults are also susceptible to this infectious dermatosis [47]. The disease characteristics were initially described in 1814 [48], but the viral background of the disease was discovered in 1905 [49]. Although the raised mollusca associated with this infection are observed to be self-limiting, lesion clearance may take from 6 months to as long as 5 years [50]. As no significant difference was observed between treated and untreated cases [51], no FDA-approved therapy exists for treatment [52]. In general, ‘active non-intervention’ is adopted as a recommended strategy in dealing with MCV infections [52]. Currently, MCV cannot be cultured in vitro, limiting the ability to investigate replication and pathogenesis [53]. Four subtypes of MCV are identified, viz., MCV-1, MCV-2, MCV-3 and MCV-4 [53]. Among these subtypes, MCV-1 causes nearly 98% of cases, particularly in children, whereas MCV-2 causes skin lesions in immunocompromised adults [53]. The double-stranded DNA genome of MCV contains 182 non-overlapping coding frames, but only half of them share homologies with other poxvirus proteins [54]. The variable region of the MCV genome hosts a number of unique genes [55]; hence, the genomic machinery of MCV is highly divergent from other mammalian chordopoxviruses [56]. Considering the unique features of MCV such as (i) restriction to humans as a significant host, (ii) a lack of a system for culture and (iii) high divergence from other poxviruses [56], continued studies of MCV are required to gain insights into viral evolution [44], pathogenesis and cellular mechanisms which control the host’s response to infection [57]. The present study focused on the genomes of MCV-1 and MCV-2 due to their higher rates of infection-causing capabilities among the four sub-types. As MCV uses humans as their natural host, long-term association with human cells may provide MCVs a platform for their own evolution [58]. In light of the fact that MCV has unique strategies to coexist with natural host [45], the present study is focused on testing the following three hypotheses to obtain insight into the co-evolving trend of the MCV genome with the host genome: (1) codon usage patterns of MCV-1 and MCV-2 are identical, (2) SCUB patterns of MCV-1 and MCV-2 slightly deviate from that of human host to avoid affecting the fitness of host, and (3) translational selection predominantly shapes the SCUB of MCV-1 and MCV-2. 

## 2. Results

### 2.1. Effect of Base Compositional Constraints on SCUB

Overall and site-specific base contents of coding sequences were estimated for MCV-1 and MCV-2 genomes to assess the effect base composition in shaping SCUB. In all selected genomes of MCV-1 and MCV-2, G and C contents were higher overall than A and T contents (Figure 1), indicating that MCV is GC-rich. In the first codon position, G content was high whereas in the second position, T content was high although overall T content was relatively low. In synonymous sites (third position), C content was high in both subtypes. Complex correlations were observed between overall and site-specific base contents in MCV-1 and MCV-2 genomes (Table 1). In both subtypes, A content was in significant negative correlation with G3, whereas A content was in positive correlation with A3 in five genomes of MC-1. In MCV-2 genomes, A and A3 were not correlated. In all genomes, T content was in significant positive correlation with A3 and T3 and was in negative correlation with C3, G3 and GC3. Except two genomes of MCV-1 and five genomes of MCV-2, other genomes exhibited significant negative correlation between G and T3 whereas positive correlation existed between G and G3 in all selected MCV I and MCV 2 genomes. In both subtypes, C content was positively correlated with C3, G3 and GC3, whereas it was negatively correlated with A3 and T3. 

### 2.2. Relative Magnitude of Selection versus Mutation

ENC and GC3 values were calculated for coding sequences of MCV-1 and MCV-2 genomes. Mean ENC values varied by 45.03 ± 0.57. Mean GC3 values were within the range of 53.308 ± 0.78. ENC values of majority of coding sequences were found to be lying in between 33–54 in MCV-1 and MCV-2 genomes indicating a clear but weak bias [59]. In the ENC vs. GC3 plot, the majority of coding sequences were lying considerably below the expected curve, indicating a high possibility of selection influencing SCUB (Figure 2). The Mann–Whitney two-sample test did not reveal any significant differences between intergenomic ENC. Moreover, a strong positive correlation between ENC and GC3 values was observed in all genomes (*p* < 0.0001), indicating the possible role of mutation as one of the major determining factors in shaping SCUB. Among the coding sequences analyzed, a few were observed to be having low SCUB (ENC ≥ 55) (Table 2). 

In the neutrality plot, strong positive correlations were observed between GC12 and GC3 in seven MCV-1 genomes (Figure 3a–g), and relatively weaker negative correlations were observed between GC12 and GC3 in two MCV-1 genomes and all selected MCV-2 genomes (Figure 3h–o). These significant correlations (*p* ≤ 0.001) indicated the critical role of mutation in shaping SCUB in the genomes of MCV-1 and MCV-2 but with varying intensities. Among the selected MCVs, in the seven genomes of MCV-1, slopes of regression lines were close to 1, revealing that mutational pressure is highly influential in determining SCUB (Figure 3a–g) [60,61], but the narrow distribution of GC3 could be due to the effect of some amount of selection. In the remaining genomes (two MCV-1 and all selected MC-2; Figure 3h–o), the scatter plots were widespread with relatively weaker correlations, and also the slopes of regression lines were ≤0.50. This indicated that mutational pressure is relatively lower and selection pressure is relatively higher in these genomes (Figure 3h–o) when compared with that of the seven MCV-1 genomes mentioned above [44]. 

PR2 bias plot revealed non-proportional usage of AT and GC count at 3rd codon position in four-fold degenerate codons in MCV-1 and MCV-2 genomes. Frequency of nucleotides A and T at degenerate positions (A3 and T3) were not equal with that of nucleotides G3 and C3 (Figure 4). AT bias at degenerate positions in the coding sequences of MCV-1 and MCV-2 deviated considerably from the center (A = T = 0.5; bias) relative to GC bias at degenerate positions in the fourfold degenerate codons.

### 2.3. Over-Represented and Under-Represented Codons

RSCU values of 59 synonymous codons of coding sequences of MCV-1 and MCV-2 were tabulated (Table 3). No strand-specific bias was observed in synonymous codon usage (Table 4). MCV-1 and MCV-2 genomes exhibited preference towards G/C ending rather than A/T ending codons in coding amino acids except methionine (Met) and tryptophan (Trp) as Met and Trp are coded by single codons. Among the thirty codons were under-represented (RSCU < 0.6), 29 were A/T ending and one was G ending (CGG for Arg). Of the twenty-one G/C ending codons over-represented (RSCU > 1.6), TTC and CAG were found to be over-represented only in MCV-2 genomes. The codon CCC was over-represented only in a single MCV-2 genome and CCG was over-represented in genomes except two MCV-1 and one MCV-2 genomes (Table 3). RSCU values of only 8 codons (~13.5%) were in the range of 0.6–1.6. Analyses of dinucleotide frequencies revealed that dinucleotide contents were not randomly distributed (χ2 test; *p* ≤ 0.05). The CC, GG and TA dinucleotides were the most under-represented in both MCV sub-types. The dinucleotides CG and GC were over-represented in all chosen MCV-1 and MCV-2 genomes. 

Among the 18 amino acids that are coded by synonymous codons, most preferred codons for six amino acids were recognized by the suboptimal isoacceptor tRNAs (GCG for Ala, CCG for Pro, ACG for Thr, TCG for Ser, CGC for Arg and ATC for Ile) in the isoacceptor tRNA pool (Table 5). Most preferred codons for remaining 12 amino acids were recognized by the abundant isoacceptor tRNAs in MCV genomes (Table 5). 

### 2.4. Major Factors Influencing SCUB

No single axis could explain majority of variations in RSCU values of coding sequences of MCV-1 and MCV-2 (Appendix A). Cumulatively, axes 1–7 accounted for more than half of the codon usage variations in both sub types of MCV. Among the seven principal axes chosen, axis 1 in MCV-1 and MCV-2 accounted for ~24% of total variations. Axis 1 was positively correlated with G3, C3, GC3 and gene length in all chosen sub types of MCV, whereas axis 1 was negatively correlated with A3, T3, ENC and CAI (Table 6). Most of the genes were spread across the axis 1 (Appendix A). Grouping of A/T ending codons to the left and G/C ending codons to the right of axis 1 was noticed in both MCV-1 and MCV-2 genomes. Cluster analyses revealed distinct grouping of MCV-1 and MCV-2 based on RSCU values (Figure 5). 

## 3. Discussion

Deciphering genomic nucleotide composition is a prerequisite for characterization of viral genomes [62]. Nucleotide composition at third codon sites is found unequal and non-random between species [63,64] and identification of major determining factors of SCUB is essential for understanding viral genome evolution [65]. In this study, patterns of SCUB and various factors which influence the formation of SCUB patterns in selected individuals of MCV-1 and MCV-2 were examined in detail. Positively correlated homogeneous base contents and negatively correlated heterogeneous base contents in MCV-1 and MCV-2 indicate the major influence of mutational pressure [66]. However, correlation analyses revealed the existence of positive heterogenous correlations (T and A3; C and G3) in all selected MC viruses. Positively correlated heterogenous correlations (T and A3; C and G3) in MCV-1 and MCV-2 revealed that natural selection by host must have influenced the SCUB patterns as in viral genomes, positive correlation between heterogeneous contents and negative correlation between homogeneous contents indicate host-induced natural selection [67]. The highest occurrence of nucleotide C at silent sites confirms the fact that overall base contents of genomes determine patterns of SCUB [33,63] as MCV genomes are GC rich [45]. 

ENC values of majority of genes were within a range (33–54), which indicates the prevalence of a distinct but weak SCUB [59]. The mean ENC value of 45.03 ± 0.57 revealed a relatively stable codon usage in genomes of MCV sub-types as ENC > 35 indicates a conserved genomic architecture [68,69]. Significant differences in intragenomic ENC (SD ≥ 5.7) and GC3 (SD ≥ 7.2) and strong positive correlation between ENC and GC3 point out the role of base compositional constraints in shaping SCUB as reported in large double-stranded DNA viruses [6,70]. Highly biased genes possess low ENC values <35 [6] indicating high levels of gene expression [71]. Variola virus, a genetically close member of MCV belonging to poxvirus group, causes a severe systemic disease with high immune response in humans, whereas MCV do not cause fulminant systemic disease and develops a low rate of immune response [45]. The low immune response developed by MCV infection can be attributed to missing of highly expressive genes of Variola virus in MCV genomic machinery which produce proteins for enabling virus–host interactions [45]. The weak SCUB (low expression) of MCV genomes can be attributed to the ability of MC viral machinery to be in the host for longer periods of time without eliciting a fulminant immune response. As the majority of genes lie far below the bell-shaped portion of the expected ENC curve, the assumption that G + C biased mutation pressure is the sole factor behind the SCUB patterns in MCV does not hold true [71]. Rejection of this null hypothesis, that is, SCUB is dictated solely by GC biased mutational pressure due to GC richness in MCV genomes reveals the possibilities of having selection influencing SCUB patterns [42] in MCV-1 and -2. The possible role of selection was further supported by the narrow distribution of GC3 in seven MCV-1 genomes and low regression slopes of remaining MCV-1 and all selected MCV-2 genomes [44]. Mean values of AT bias [A3/(A3 + T3)] and GC [G3/(G3 + C3)] bias were greater than 0.5, indicating preference of purines over pyrimidines, that is, A over T and G over C [42,72] in synonymous codons of four-fold degenerate amino acids. 

The strong preference towards G/C ending codons was due to over-representation of CG/GC dinucleotides in MCV genomes. The low frequency of GG dinucleotide resulted in the under-representation of CGG codon in coding amino acid Arg. This confirms the fact that bias in dinucleotide frequencies shape SCUB [6,73]. The under-representation of TA dinucleotide in MCV genomes may possibly be due to low thermal stability [74] resulting in destabilization of mRNA coupled with sensitivity of uracil in UpA (uracil-phosphate-adenine) to cytoplasmic RNase [75] to regulate mRNA turnover in a cell [42]. Among the GC containing codons, GCG, CCG, CGC, TCG and ACG were used preferentially (RSCU > 1.5) whereas CGA, CGG and CGT were under-represented (RSCU < 0.6). The low frequencies of GG and GT dinucleotides can justify the under-representation of CGG and CGT. The possible reason for the low preference of CGA may be attributed to the low overall A content. These results suggest that SCUB in MCV genomes is largely influenced by dinucleotide bias as reported [42,76]. Although codon usage patterns shared some common features as mentioned above, the cluster analysis (Figure 5) revealed a clear difference in RSCU patterns of MCV 1 and MCV 2, as both sub-types formed distinct clusters. 

Role of translation selection in shaping SCUB in MCV can be confirmed by checking whether most preferred codons are recognized by most abundant isoacceptor tRNAs in the isoacceptor tRNA pool [9,42]. In the selected MCV sub-types, most preferred codons of 12 amino acids correspond to the most abundant isoacceptor tRNAs, indicating the role of translational selection [77,78]. Most of the non-optimal codon–anticodon base pairing occurred with CG dinucleotide containing codons (GCG for Ala, CCG for Pro, ACG for Thr, TCG for Ser, CGC) in MCV genomes, that is, most preferred CG dinucleotide containing codons in MCV were translated by rare tRNAs. This can be considered as a selective force to keep a low rate of translation [79,80] in the beginning to develop proper folding of viral proteins [81] for evading host immunity [82] by reducing the anti-viral response from the host [73]. Moreover, strong positive correlations between CAI and ENC (*p* < 0.0001) also indicate selection pressure as observed in Nipah viruses [42] as correlation between ENC and CAI determine the relative magnitude selection versus mutation [83]. The strong correlations between axis 1 and silent base contents (A3, T3, G3 and C3) pointed out the relative influence of mutational pressure due to compositional constraints in shaping SCUB. CAI values are associated with selection and ENC values reveals SCUB which can be due to either mutation/selection [42]. The strong correlation between axis 1 and these two indices (ENC and CAI) specified the relative high magnitude of selection over mutation in MCV genomes. 

Similar to the pattern observed in MCV genomes, host cells also used G/C ending codons most preferentially [81,84]. Although both MCV and host cells preferred G/C ending codons, the non-optimal codon-anticodon base pairing of most preferred codons containing CG dinucleotides indicated that MCV genomes may follow a deliberate slight deviation from host codon usage to remain in the host for a certain period to become adapted to host for acquiring ambient ‘climate’ for genome evolution [39]. Viral adaptation to host in terms of codon usage is essential for the infection to be successful in human host [41] either due to coevolution of human genome along with infected viral genome or due to human genome evolution from viral genome [85]. 

## 4. Conclusions

This study was performed to test the veracity of following three hypotheses. 

First hypothesis—Codon usage patterns of MCV-1 and MCV-2 are identical: Although SCUB patterns of MCV-1 and MCV-2 shared common features, apparent intrinsic differences existed in codon usage patterns as revealed by grouping of MCV-1 and MCV-2 in cluster analysis. Thus, the first hypothesis was not accepted.

Second hypothesis—SCUB patterns of MCV 1 and MCV 2 slightly deviate from that of human host to avoid affecting the fitness of host: Despite both human and MCV genomes used G/C ending codons, most preferred codons containing CG dinucleotides were not recognized by most abundant isoacceptor isotypes. This indicated that MCV genomes followed a slight deviation from codon usage pattern of host cells. Thus, the second hypothesis was accepted. 

Third hypothesis—Translational selection predominantly shapes the SCUB of MCV-1 and MCV-2: The findings such as strong correlations between ENC and CAI, strong correlation between axis 1 and ENC and axis 1 and CAI, recognition of majority of most preferred codons in MCV genomes by the most abundant isoacceptor isotypes in host cells indicates dominant role of selection along with mutational pressure. Thus, the third hypothesis was also accepted. 

## 5. Materials and Methods

### 5.1. Data Retrieval

The coding sequences (CDS) with exact initiation and termination codons of nine MCV-1 and six MCV-2 genomes were retrieved in FASTA format from GenBank database of the National Center for Biotechnology Information (NCBI). Details such as subtypes, accession numbers, country of isolation, total number of CDS, selected CDS and size of genomes are provided in Table 7. Only coding sequences of length ≥ 300 nucleotides were selected for analyses to avoid sampling errors and stochastic variations [6]. Sequences were aligned using MUSCLE algorithm [86] embedded in MEGA X [87]. For each genome, coding sequences on the plus and minus strands were grouped separately to assess strand-specific codon usage bias.

### 5.2. Relative Synonymous Codon Usage

Relative synonymous codon usage (RSCU) is an important measure to analyze the biased usage of synonymous codons in coding a given amino acid [88]. RSCU value of a codon which codes for a given amino acid is calculated as the ratio of observed occurrences of that codon to the expected occurrences of the same codon provided all synonymous codons of that particular amino acid are used equally [27]. If RSCU value of a codon is greater than 1, it indicates preferred usage over its synonymous counterparts [27,89]. If RSCU value is less than 1, it indicates non-preferred usage and for rare codons, RSCU values fall below 0.66 [32]. No bias is indicated if RSCU value is 1 [27]. RSCU value was calculated according to the equation given below [27]
$${RSCU}_{mn}\;=\;\frac{F_{mn}}{\sum_{m}^{ci}F_{mn}}\times ci$$
where, *RSCU_mn_* is the relative synonymous codon usage value of mth codon of nth amino acid. *F_mn_* is the observed frequency of mth codon of nth amino acid and *ci* is the number of standard synonymous codons of nth amino acid, i.e., level of codon degeneracy.

### 5.3. Dinucleotide Analysis

Dinucleotide frequencies were estimated to check whether any dinucleotides from possible 16 combinations are preferably used as dinucleotide bias is linked with SCUB [33]. Dinucleotide frequency was calculated as follows [42]
$$P_{xy}=\frac{F_{xy}}{F_{x}F_{y}}$$
where *F_x_* = frequency of nucleotide x, *F_y_* = frequency of nucleotide y and *F_xy_* is the frequency of dinucleotide xy. The odds ratio is defined as the ratio of observed frequency of a dinucleotide to the expected frequency of that particular dinucleotide. If odds ratio of a given dinucleotide falls above 1.25, it is a sign of over-representation and if the value falls below 0.78, it is a sign of under-representation [42,76].

### 5.4. ENC vs. GC3 Plot

Effective number of codons (*ENC*) was calculated to assess the extent of SCUB. ENC values range from 20 (extreme bias of synonymous codon usage, i.e., one codon for one amino acid) to 61 (near uniform synonymous codon usage). Expected *ENC* value of a given sequence is calculated as follows [71]
$$ENC=2+s+\frac{29}{s^{2}+{\left(1-s\right)}^{2}}$$
where *s* = GC content at the synonymous position of codons (GC3).

In ENC vs. GC3 plot, expected curve is a bell-shaped curve indicating the expected values of ENC (ordinate) determined solely by base composition (GC3; abscissa) as per the equation above [71]. In the biological system, for a given sequence, observed ENC values may not always follow the path of expected curve. If observed *ENC* values fall on or just near the expected curve, it can be assumed that compositional constraints influence the SCUB to a great extent [89]. On the other hand, if observed ENC values fall considerably below the expected curve, it can be assumed that certain other factors (for, e.g., selection) must be influencing the shaping of SCUB [89]. Coding sequences having ENC values ≤ 30 are considered to be highly biased and those with ENC values ≥ 55 are considered to be less biased [59].

### 5.5. Neutrality Plot

Average GC composition at 1st, 2nd and 3rd codon position were calculated. Using GC values at 1st and 2nd positions (GC1 + GC2 = GC12; ordinate) and GC3 (abscissa), neutrality plot was developed to assess the mutation—selection balance in framing SCUB [44]. In the scatter plot, each CDS is indicated by a dot and existence of high correlation between GC12 and GC3 with slope coefficient close to 1 indicates the role of mutation in shaping SCUB [90]. If dots are widespread with no correlation between GC3 and GC12 with slope coefficient tends towards 0, selection is presumed to be possibly influencing the SCUB [6,44]. 

### 5.6. Parity Rule 2 Plot

Parity rule 2 (PR2) plot was developed to determine relative magnitude of mutation and selection in framing base composition of coding sequences [44]. In this plot, AT bias [A/(A + T)] and GC bias [G/(G + C)] are plotted on ordinate and abscissa [91]. If equal proportion of nucleotides (A = T = G = C = 0.25) is assumed, 0.5 would be the value at the center of the plot indicating that effects of mutation and selection are equal [92]. In this study, AT and GC bias at the third codon positions [A3/(A3 + T3), G3/(G3 + C3)] of four-fold degenerate amino acids of each coding sequence were plotted as PR2 biases at the synonymous positions are relatively more significant [93,94].

### 5.7. Correspondence Analysis

Correspondence analyses (CA) was performed on 59 synonymous codons (excluding ATG for Met, TGG for Trp, termination codons TAA, TAG and TGA) by assuming each coding sequence as a 59-dimensional vector with each dimension identical to RSCU value of a codon [61,95] for delineating SCU variations across the genes of MCV genomes. The relative importance of each codon over each orthogonal axis is represented by eigen value [96]. The total variation of codon usage was partitioned across 59 orthogonal axes in terms of percentage variation accounted by each CA-axis [97]. The first axis of CA explained majority of variations followed by subsequent axes holding a declining number of variations [97]. The number of axes for spearman’s rank correlation analyses to study the relative influence of various factors on SCUB was determined based on the condition that selected axes account for majority (>50%) of codon usage variations. 

### 5.8. Cluster Analysis

Cluster analysis was performed on the pooled RSCU values of coding sequences of MCV 1 and MCV 2 genomes to study the pattern of codon usage in subtypes of selected MCV based on grouping of subtypes in terms of codon usage [6,70]. A 15 × 59 matrix was constructed in which rows corresponded to 15 MCV strains (nine MCV 1 and six MCV 2) and columns corresponded to pooled RSCU values of 59 codons. The method employed for clustering MCV 1 and MCV 2 subtypes based on RSCU values was unweighted pair-group average clustering based on Euclidean distances [6].

### 5.9. Statistical Analysis and the Softwares Used

Dambe ver 7.3.2 [98] was employed to compute overall base contents, site-specific nucleotide compositions, RSCU, ENC and codon adaptation index (CAI) values. Isoacceptor tRNA pool was identified using an online tool (GtRNAdb: Genomic tRNA database) [42]. All correlation analyses were carried out using non-parametric Spearman rank correlation method [6,97]. Non-parametric Spearman rank correlation method, Mann–Whitney 2-sample test and cluster analysis were performed using PAST 4.03 [99]. For all statistical analyses, the level of significance was taken as *p* < 0.05.

## Figures and Tables

**Figure 1 pathogens-10-01649-f001:**
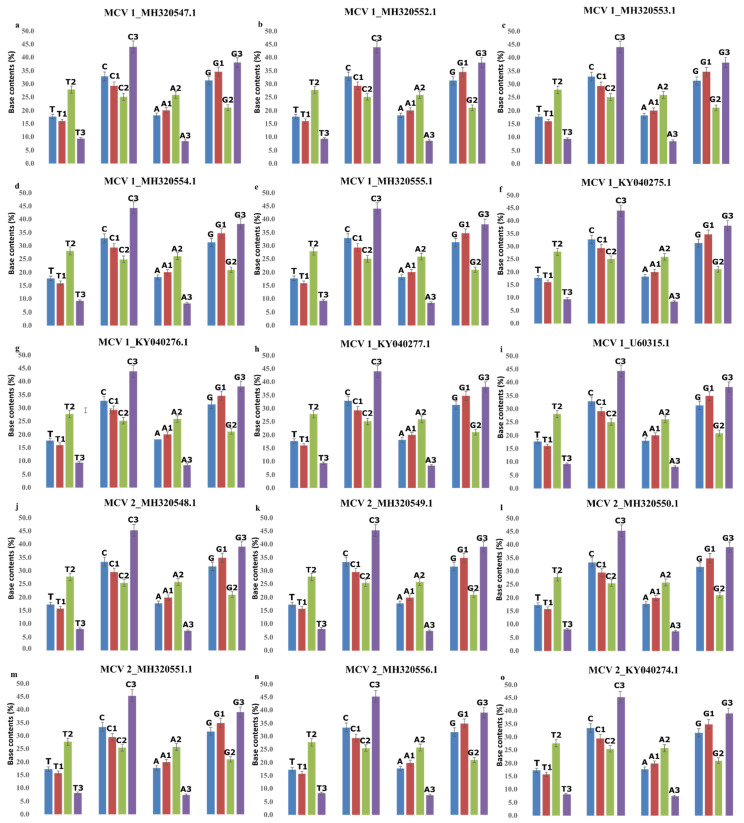
Nucleotide composition: Overall and site-specific base composition of selected coding sequences in MCV-1 and MCV-2, (**a**) MH320547.1 (MCV 1), (**b**) MH320552.1 (MCV 1), (**c**) MH320553.1 (MCV 1), (**d**) MH320554.1 (MCV 1), (**e**) MH320555.1 (MCV 1), (**f**) KY040275.1 (MCV 1), (**g**) KY040276.1 (MCV 1), (**h**) KY040277.1 (MCV 1), (**i**) U60315.1 (MCV 1), (**j**) MH320548.1 (MCV 2), (**k**) MH320549.1 (MCV 2), (**l**) MH320550.1 (MCV 2), (**m**) MH320551.1 (MCV 2), (**n**) MH320556.1 (MCV 2) and (**o**) KY040274.1 (MCV 2).

**Figure 2 pathogens-10-01649-f002:**
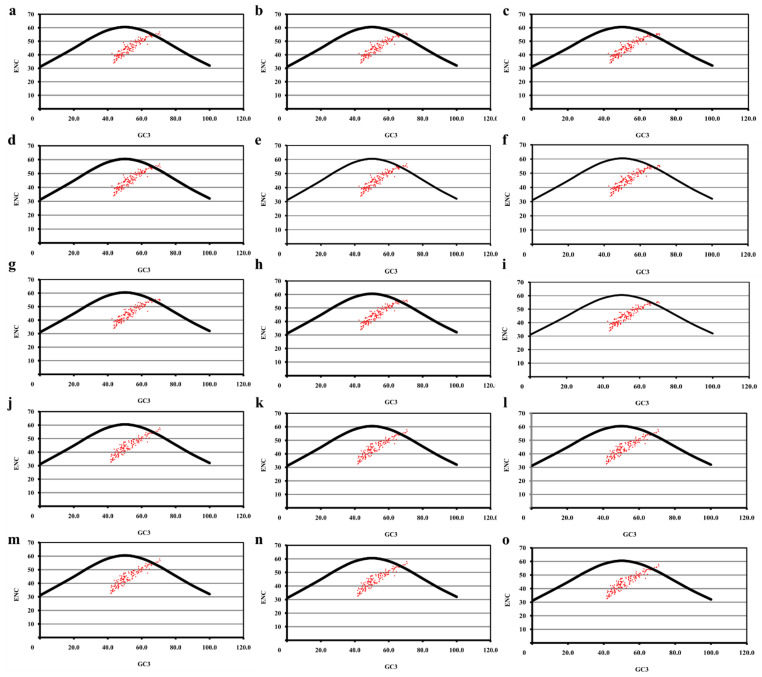
Relative magnitude selection vs. mutation. ENC vs. GC3 plots of (**a**) MH320547.1 (MCV 1), (**b**) MH320552.1 (MCV 1), (**c**) MH320553.1 (MCV 1), (**d**) MH320554.1 (MCV 1), (**e**) MH320555.1 (MCV 1), (**f**) KY040275.1 (MCV 1), (**g**) KY040276.1 (MCV 1), (**h**) KY040277.1 (MCV 1), (**i**) U60315.1 (MCV 1), (**j**) MH320548.1 (MCV 2), (**k**) MH320549.1 (MCV 2), (**l**) MH320550.1 (MCV 2), (**m**) MH320551.1 (MCV 2), (**n**) MH320556.1 (MCV 2) and (**o**) KY040274.1 (MCV 2).

**Figure 3 pathogens-10-01649-f003:**
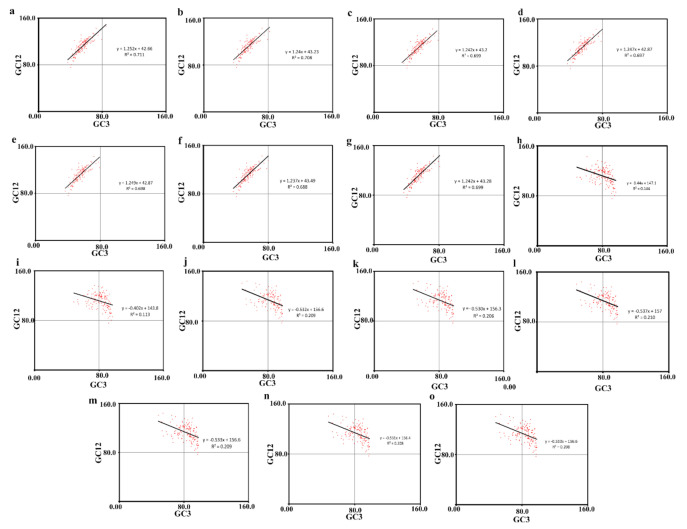
GC composition and codon bias in MCV genomes. Neutrality plots of (**a**) MH320547.1 (MCV 1), (**b**) MH320552.1 (MCV 1), (**c**) MH320553.1 (MCV 1), (**d**) MH320554.1 (MCV 1), (**e**) MH320555.1 (MCV 1), (**f**) KY040275.1 (MCV 1), (**g**) KY040276.1 (MCV 1), (**h**) KY040277.1 (MCV 1), (**i**) U60315.1 (MCV 1), (**j**) MH320548.1 (MCV 2), (**k**) MH320549.1 (MCV 2), (**l**) MH320550.1 (MCV 2), (**m**) MH320551.1 (MCV 2), (**n**) MH320556.1 (MCV 2) and (**o**) KY040274.1 (MCV 2).

**Figure 4 pathogens-10-01649-f004:**
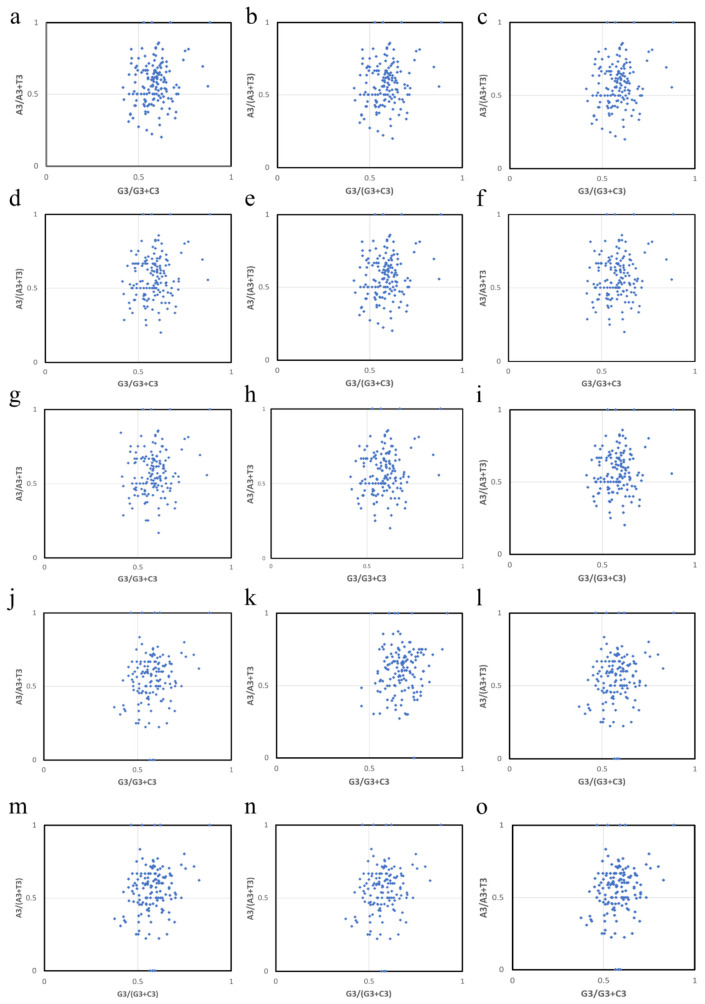
Deviation from equal usage of nucleotides at 3rd codon position in 4-fold degenerate amino acids. Parity rule 2 plots of (**a**) MH320547.1 (MCV 1), (**b**) MH320552.1 (MCV 1), (**c**) MH320553.1 (MCV 1), (**d**) MH320554.1 (MCV 1), (**e**) MH320555.1 (MCV 1), (**f**) KY040275.1 (MCV 1), (**g**) KY040276.1 (MCV 1), (**h**) KY040277.1 (MCV 1), (**i**) U60315.1 (MCV 1), (**j**) MH320548.1 (MCV 2), (**k**) MH320549.1 (MCV 2), (**l**) MH320550.1 (MCV 2), (**m**) MH320551.1 (MCV 2), (**n**) MH320556.1 (MCV 2) and (**o**) KY040274.1 (MCV 2).

**Figure 5 pathogens-10-01649-f005:**
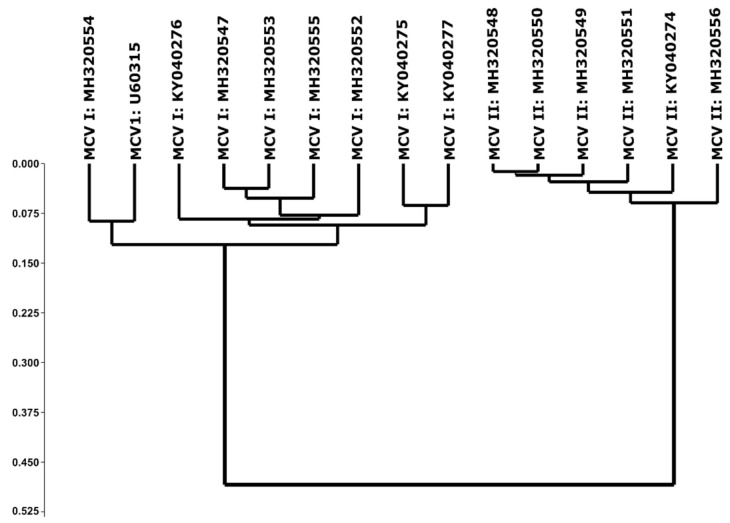
Distinctive codon bias patterns of MCV 1 and MCV 2. Cluster analysis of RSCU values of MH320547.1 (MCV 1), MH320552.1 (MCV 1), MH320553.1 (MCV 1), MH320554.1 (MCV 1), MH320555.1 (MCV 1), KY040275.1 (MCV 1), KY040276.1 (MCV 1), KY040277.1 (MCV 1), U60315.1 (MCV 1), MH320548.1 (MCV 2), MH320549.1 (MCV 2), MH320550.1 (MCV 2), MH320551.1 (MCV 2), MH320556.1 (MCV 2) and KY040274.1 (MCV 2).

**Table 1 pathogens-10-01649-t001:** Correlation analysis between silent and overall base contents.

Strains	Nucleotides	A3	T3	G3	C3	GC3
MH320547.1 (MCV 1)	A	0.1604	0.0926	−0.3491 *	−0.0331	−0.1356
	T	0.2537 *	0.5157 *	−0.3821 *	−0.3033 *	−0.3885 *
	G	−0.1000	−0.1767 *	0.5699 *	−0.1169	0.1308
	C	−0.4028 *	−0.5017 *	0.2485 *	0.4974 *	0.4751 *
	GC	−0.2550 *	−0.3593 *	0.4949 *	0.1779 *	0.3117 *
MH320552.1 (MCV 1)	A	0.1736 *	0.1081	−0.3694 *	−0.0368	−0.1476
	T	0.2449 *	0.5064 *	−0.3947 *	−0.2822 *	−0.3801 *
	G	−0.0990	−0.1780 *	0.5822 *	−0.1244	0.1295
	C	−0.3965 *	−0.4999 *	0.2527 *	0.4807 *	0.4699 *
	GC	−0.2488 *	−0.3543 *	0.5070 *	0.1592	0.3046 *
MH320553.1 (MCV 1)	A	0.1660 *	0.0940	−0.3628 *	−0.0289	−0.1384
	T	0.2566 *	0.5195 *	−0.4011 *	−0.2958 *	−0.3921 *
	G	−0.0988	−0.1794 *	0.5730 *	−0.1207	0.1305
	C	−0.4042 *	−0.5021 *	0.2691 *	0.4865 *	0.4762 *
	GC	−0.2566 *	−0.3609 *	0.5122 *	0.1693 *	0.3126 *
MH320554.1 (MCV 1)	A	0.1552	0.1029	−0.3594 *	−0.0252	−0.1297
	T	0.2714 *	0.5201 *	−0.4094 *	−0.2966 *	−0.4028 *
	G	−0.1032	−0.1827 *	0.5837 *	−0.1227	0.1313
	C	−0.4048 *	−0.5195 *	0.2492 *	0.4985 *	0.4814 *
	GC	−0.2527 *	−0.3673 *	0.5103 *	0.1658 *	0.3101 *
MH320555.1 (MCV 1)	A	0.1658 *	0.0921	−0.3558 *	−0.0271	−0.1348
	T	0.2545 *	0.5172 *	−0.4041 *	−0.2928 *	−0.3910 *
	G	−0.1031	−0.1827 *	0.5755 *	−0.1164	0.1346
	C	−0.4064 *	−0.5042 *	0.2694 *	0.4876 *	0.4768 *
	GC	−0.2606 *	−0.3633 *	0.5153 *	0.1710 *	0.3154 *
KY040275.1 (MCV 1)	A	0.1703 *	0.0996	−0.3683 *	−0.0318	−0.1419
	T	0.2555 *	0.5285 *	−0.3915 *	−0.3024 *	−0.3953 *
	G	−0.0968	−0.1763 *	0.5635 *	−0.1171	0.1277
	C	−0.4203 *	−0.5211 *	0.2786 *	0.5014 *	0.4943 *
	GC	−0.2657 *	−0.3724 *	0.5131 *	0.1813 *	0.3237 *
KY040276.1 (MCV 1)	A	0.1727 *	0.1026	−0.3726 *	−0.0357	−0.1441
	T	0.2622 *	0.5162 *	−0.4032 *	−0.2969 *	−0.3981 *
	G	−0.0939	−0.1687 *	0.5673 *	−0.1213	0.1242
	C	−0.4041 *	−0.5081 *	0.2739 *	0.4861 *	0.4778 *
	GC	−0.2613 *	−0.3638 *	0.5191 *	0.1736 *	0.3182 *
KY040277.1 (MCV 1)	A	0.1473	0.0913	−0.3624 *	−0.0117	−0.1184
	T	0.3011 *	0.5498 *	−0.4200 *	−0.3347 *	−0.4341 *
	G	−0.0797	−0.1580	0.5582 *	−0.1386	0.1068
	C	−0.4010 *	−0.5109 *	0.2610 *	0.4856 *	0.4740 *
	GC	−0.2469 *	−0.3594 *	0.5061 *	0.1603	0.3026 *
U60315.1 (MCV 1)	A	0.0936	0.0630	−0.3678 *	0.0505	−0.0686
	T	0.3099 *	0.5491 *	−0.4489 *	−0.3349 *	−0.4462 *
	G	−0.0664	−0.1345	0.5817 *	−0.1701 *	0.0932
	C	−0.3953 *	−0.5171 *	0.2658 *	0.4852 *	0.4757 *
	GC	−0.2314 *	−0.3414 *	0.5288 *	0.1344	0.2914 *
MH320548.1 (MCV 2)	A	0.0950	0.0608	−0.3298 *	0.0060	−0.0775
	T	0.2825 *	0.5001 *	−0.4136 *	−0.3005 *	−0.4072 *
	G	−0.0615	−0.1576	0.5516 *	−0.1242	0.1174
	C	−0.3580 *	−0.3924 *	0.2072 *	0.4224 *	0.3836 *
	GC	−0.2178 *	−0.3177 *	0.4759 *	0.1525	0.2764 *
MH320549.1(MCV 2)	A	0.0920	0.0603	−0.3328 *	0.0078	−0.0744
	T	0.2815 *	0.5072 *	−0.4193 *	−0.3040 *	−0.4110 *
	G	−0.0603	−0.1627	0.5520 *	−0.1178	0.1191
	C	−0.3547 *	−0.3926 *	0.2128 *	0.4191 *	0.3820 *
	GC	−0.2167 *	−0.3251 *	0.4814 *	0.1577	0.2798 *
MH320550.1 (MCV 2)	A	0.0934	0.0611	−0.3329 *	0.0102	−0.0761
	T	0.2799 *	0.4978 *	−0.4169 *	−0.2942 *	−0.4036 *
	G	−0.0574	−0.1538	0.5518 *	−0.1305	0.1125
	C	−0.3524 *	−0.3901 *	0.2057 *	0.4170 *	0.3794 *
	GC	−0.2150 *	−0.3160 *	0.4778 *	0.1465	0.2729 *
MH320551.1 (MCV 2)	A	0.0935	0.0595	−0.3317 *	0.0101	−0.0759
	T	0.2820 *	0.4971 *	−0.4043 *	−0.3042 *	−0.4064 *
	G	−0.0535	−0.1489	0.5435 *	−0.1279	0.1099
	C	−0.3564 *	−0.3900 *	0.2058 *	0.4182 *	0.3818 *
	GC	−0.2132 *	−0.3162 *	0.4715 *	0.1535	0.2757 *
MH320556.1 (MCV 2)	A	0.0966	0.0807	−0.3305 *	0.0061	−0.0773
	T	0.2825 *	0.5075 *	−0.4165 *	−0.3041 *	−0.4089 *
	G	−0.0578	−0.1764 *	0.5481 *	−0.1235	0.1143
	C	−0.3574 *	−0.3973 *	0.2066 *	0.4209 *	0.3822 *
	GC	−0.2183 *	−0.3417 *	0.4778 *	0.1552	0.2778 *
KY040274.1 (MCV 2)	A	0.0978	0.0639	−0.3291 *	0.0016	−0.0809
	T	0.2813 *	0.5002 *	−0.4067 *	−0.3019 *	−0.4054 *
	G	−0.0595	−0.1554	0.5480 *	−0.1264	0.1148
	C	−0.3520 *	−0.3928 *	0.1966 *	0.4247 *	0.3810 *
	GC	−0.2186 *	−0.3223 *	0.4705 *	0.1569	0.2785 *

* level of significance was measured at *p* < 0.05.

**Table 2 pathogens-10-01649-t002:** Gene sequences with high ENC ≥ 55.

Subtype	Accession ID	Genes
MCV 1	MH320547.1	150R, 133L, 054L, 148R, 132L, 152R
MCV 1	MH320552.1	133L, 054L, 148R, 132L, 152R
MCV 1	MH320553.1	150R, 133L, 054L, 148R, 132L, 152R
MCV 1	MH320554.1	133L, 148R, 152R, 156R, 055R
MCV 1	MH320555.1	150R, 133L, 054L, 148R, 132L, 152R, 156R
MCV 1	KY040275.1	133L, 055R, 152.1R, 148R, 132L
MCV 1	KY040276.1	133L, 152.1R, 148R,132L,054L
MCV 1	KY040277.1	133L, 152.1R, 148R, 132L, 054L
MCV 1	U60315.1	133L, 055R, 148R, 132L
MCV 2	MH320548.1	152.1R, 148R, 151L, 012L, 010R
MCV 2	MH320549.1	151L, 010R, 148R,145.1R, 152.1R, 012L
MCV 2	MH320550.1	151L, 010R, 148R, 152.1R, 012L
MCV 2	MH320551.1	151L, 010R, 148R, 152.1R, 012L
MCV 2	MH320556.1	151L, 010R, 148R, 152.1R, 012L
MCV 2	KY040274.1	151L, 010R, 148R, 152.1R, 012L

**Table 3 pathogens-10-01649-t003:** Overall relative synonymous codon usage in the selected MCV genomes.

Codon	AA	1	2	3	4	5	6	7	8	9	10	11	12	13	14	15
GCT	A	0.322	0.317	0.322	0.314	0.327	0.313	0.315	0.314	0.311	0.287	0.287	0.286	0.287	0.284	0.282
GCG	A	2.052	2.080	2.058	2.076	2.060	2.067	2.075	2.072	2.079	2.160	2.161	2.161	2.157	2.158	2.167
GCC	A	1.098	1.085	1.093	1.112	1.093	1.098	1.086	1.094	1.105	1.128	1.128	1.126	1.131	1.131	1.125
GCA	A	0.528	0.519	0.528	0.498	0.520	0.521	0.524	0.520	0.505	0.425	0.425	0.427	0.425	0.427	0.425
TGT	C	0.377	0.372	0.372	0.353	0.357	0.379	0.386	0.376	0.365	0.330	0.328	0.329	0.330	0.325	0.329
TGC	C	1.623	1.628	1.628	1.647	1.643	1.621	1.614	1.624	1.635	1.670	1.672	1.671	1.670	1.675	1.671
GAT	D	0.244	0.242	0.244	0.241	0.246	0.241	0.250	0.240	0.236	0.220	0.219	0.219	0.219	0.216	0.218
GAC	D	1.756	1.758	1.756	1.759	1.754	1.759	1.750	1.760	1.764	1.780	1.781	1.781	1.781	1.784	1.782
GAG	E	1.601	1.601	1.602	1.609	1.601	1.606	1.604	1.607	1.615	1.637	1.637	1.637	1.639	1.633	1.637
GAA	E	0.399	0.399	0.398	0.391	0.399	0.394	0.396	0.393	0.385	0.363	0.363	0.363	0.361	0.367	0.363
TTT	F	0.458	0.458	0.454	0.448	0.447	0.468	0.465	0.466	0.456	0.382	0.383	0.382	0.383	0.385	0.382
TTC	F	1.542	1.542	1.546	1.552	1.553	1.532	1.535	1.534	1.544	1.618	1.617	1.618	1.617	1.615	1.618
GGT	G	0.298	0.302	0.299	0.289	0.291	0.307	0.308	0.307	0.304	0.271	0.271	0.271	0.271	0.272	0.272
GGG	G	0.914	0.913	0.921	0.915	0.914	0.917	0.911	0.929	0.906	0.835	0.833	0.835	0.833	0.832	0.837
GGC	G	2.243	2.246	2.240	2.259	2.246	2.231	2.234	2.233	2.247	2.366	2.366	2.366	2.369	2.369	2.363
GGA	G	0.545	0.539	0.540	0.537	0.549	0.545	0.547	0.532	0.543	0.527	0.529	0.527	0.527	0.527	0.529
CAC	H	1.722	1.724	1.722	1.718	1.718	1.718	1.724	1.721	1.720	1.774	1.774	1.770	1.774	1.773	1.774
CAT	H	0.278	0.276	0.278	0.282	0.282	0.282	0.276	0.279	0.280	0.226	0.226	0.230	0.226	0.227	0.226
ATT	I	0.561	0.563	0.563	0.556	0.557	0.561	0.565	0.564	0.563	0.426	0.425	0.426	0.426	0.440	0.426
ATA	I	0.156	0.160	0.158	0.160	0.158	0.162	0.155	0.154	0.147	0.151	0.152	0.151	0.151	0.155	0.151
ATC	I	2.283	2.277	2.280	2.284	2.285	2.276	2.280	2.282	2.290	2.424	2.423	2.424	2.424	2.404	2.424
AAA	K	0.336	0.338	0.330	0.309	0.327	0.323	0.325	0.321	0.309	0.265	0.263	0.264	0.264	0.265	0.262
AAG	K	1.664	1.662	1.670	1.691	1.673	1.677	1.675	1.679	1.691	1.735	1.737	1.736	1.736	1.735	1.738
CTA	L	0.298	0.294	0.297	0.282	0.295	0.293	0.302	0.296	0.285	0.232	0.232	0.233	0.233	0.235	0.232
CTC	L	1.175	1.173	1.175	1.173	1.177	1.176	1.167	1.176	1.177	1.208	1.203	1.208	1.203	1.206	1.210
CTG	L	2.193	2.204	2.195	2.219	2.199	2.197	2.194	2.198	2.217	2.285	2.287	2.284	2.291	2.282	2.284
CTT	L	0.334	0.329	0.333	0.327	0.329	0.335	0.337	0.330	0.321	0.275	0.277	0.274	0.273	0.277	0.275
TTA	L	0.265	0.272	0.265	0.268	0.265	0.265	0.267	0.261	0.269	0.301	0.301	0.301	0.302	0.302	0.301
TTG	L	1.735	1.728	1.735	1.732	1.735	1.735	1.733	1.739	1.731	1.699	1.699	1.699	1.698	1.698	1.699
AAC	N	1.740	1.739	1.738	1.740	1.736	1.742	1.734	1.744	1.749	1.775	1.774	1.775	1.775	1.781	1.775
AAT	N	0.260	0.261	0.262	0.260	0.264	0.258	0.266	0.256	0.251	0.225	0.226	0.225	0.225	0.219	0.225
CCA	P	0.379	0.384	0.378	0.365	0.370	0.387	0.377	0.388	0.375	0.331	0.332	0.329	0.331	0.339	0.328
CCC	P	1.525	1.530	1.534	1.545	1.530	1.525	1.518	1.528	1.541	1.591	1.595	1.595	1.592	1.597	1.605
CCT	P	0.482	0.485	0.483	0.489	0.487	0.496	0.483	0.484	0.487	0.456	0.453	0.457	0.450	0.465	0.456
CCG	P	1.614	1.601	1.604	1.602	1.612	1.593	1.622	1.601	1.596	1.622	1.621	1.620	1.627	1.599	1.611
CAA	Q	0.435	0.439	0.437	0.429	0.432	0.438	0.434	0.439	0.422	0.390	0.393	0.391	0.390	0.393	0.389
CAG	Q	1.565	1.561	1.563	1.571	1.568	1.562	1.566	1.561	1.578	1.610	1.607	1.609	1.610	1.607	1.611
AGA	R	0.725	0.709	0.723	0.694	0.729	0.707	0.718	0.726	0.699	0.759	0.757	0.761	0.763	0.742	0.742
AGG	R	1.275	1.291	1.277	1.306	1.271	1.293	1.282	1.274	1.301	1.241	1.243	1.239	1.237	1.258	1.258
CGA	R	0.261	0.272	0.265	0.255	0.267	0.267	0.261	0.265	0.250	0.244	0.243	0.244	0.242	0.244	0.252
CGC	R	2.837	2.825	2.834	2.868	2.841	2.826	2.831	2.830	2.877	2.889	2.892	2.887	2.895	2.879	2.882
CGG	R	0.543	0.537	0.543	0.531	0.537	0.537	0.551	0.534	0.532	0.547	0.545	0.550	0.543	0.547	0.546
CGT	R	0.359	0.366	0.358	0.346	0.354	0.371	0.357	0.371	0.340	0.320	0.320	0.320	0.320	0.330	0.319
AGC	S	1.670	1.660	1.671	1.670	1.670	1.669	1.674	1.667	1.679	1.721	1.722	1.721	1.721	1.707	1.721
AGT	S	0.330	0.340	0.329	0.330	0.330	0.331	0.326	0.333	0.321	0.279	0.278	0.279	0.279	0.293	0.279
TCA	S	0.220	0.230	0.220	0.230	0.220	0.236	0.228	0.228	0.217	0.188	0.190	0.188	0.189	0.201	0.187
TCC	S	1.489	1.462	1.487	1.493	1.487	1.486	1.490	1.484	1.510	1.534	1.536	1.532	1.533	1.526	1.542
TCG	S	1.752	1.758	1.750	1.753	1.750	1.751	1.747	1.755	1.743	1.798	1.794	1.800	1.797	1.794	1.789
TCT	S	0.539	0.550	0.543	0.525	0.544	0.528	0.535	0.533	0.530	0.480	0.480	0.480	0.481	0.479	0.481
ACC	T	1.161	1.157	1.163	1.167	1.162	1.156	1.148	1.157	1.186	1.221	1.218	1.225	1.213	1.217	1.221
ACA	T	0.519	0.523	0.516	0.512	0.509	0.534	0.533	0.529	0.494	0.425	0.425	0.422	0.426	0.440	0.437
ACG	T	1.902	1.907	1.905	1.911	1.909	1.900	1.897	1.900	1.919	1.955	1.957	1.957	1.962	1.948	1.948
ACT	T	0.418	0.413	0.416	0.410	0.420	0.410	0.421	0.413	0.401	0.399	0.400	0.396	0.400	0.395	0.395
GTT	V	0.293	0.290	0.291	0.280	0.286	0.301	0.312	0.293	0.278	0.259	0.263	0.258	0.259	0.259	0.258
GTG	V	2.328	2.339	2.329	2.336	2.328	2.335	2.332	2.339	2.345	2.453	2.453	2.451	2.456	2.462	2.455
GTC	V	1.066	1.061	1.065	1.070	1.070	1.056	1.042	1.053	1.070	1.049	1.046	1.049	1.051	1.047	1.046
GTA	V	0.313	0.310	0.314	0.314	0.316	0.307	0.314	0.315	0.307	0.240	0.239	0.242	0.234	0.232	0.241
TAC	Y	1.777	1.778	1.777	1.773	1.777	1.773	1.774	1.774	1.775	1.786	1.788	1.786	1.786	1.789	1.788
TAT	Y	0.223	0.222	0.223	0.227	0.223	0.227	0.226	0.226	0.225	0.214	0.212	0.214	0.214	0.211	0.212

1. MH320547.1 (MCV 1) 2. MH320552.1 (MCV 1) 3. MH320553.1 (MCV 1) 4. MH320554.1 (MCV 1) 5. MH320555.1 (MCV 1) 6. KY040275.1 (MCV 1) 7. KY040276.1 (MCV 1) 8. KY040277.1 (MCV 1) 9. U60315.1 (MCV 1) 10. MH320548.1 (MCV 2) 11. MH320549.1 (MCV 2) 12. MH320550.1 (MCV 2) 13. MH320551.1 (MCV 2) 14. MH320556.1 (MCV 2) 15. KY040274.1 (MCV 2).

**Table 4 pathogens-10-01649-t004:** Strand-specific codon usage in the selected MCV genomes.

Codons	AA	1		2		3		4		5		6		7		8		9		10		11		12		13		14		15	
		+	−	+	−	+	−	+	−	+	−	+	−	+	−	+	−	+	−	+	−	+	−	+	−	+	−	+	−	+	−
GCT	A	0.333	0.313	0.326	0.309	0.334	0.310	0.343	0.288	0.346	0.310	0.327	0.301	0.330	0.302	0.326	0.304	0.325	0.299	0.303	0.274	0.304	0.274	0.303	0.273	0.304	0.274	0.299	0.273	0.301	0.267
GCG	A	2.026	2.075	2.051	2.104	2.012	2.098	2.031	2.116	2.014	2.100	2.036	2.095	2.056	2.091	2.047	2.093	2.047	2.107	2.159	2.161	2.157	2.164	2.157	2.164	2.150	2.162	2.149	2.165	2.170	2.165
GCC	A	1.086	1.107	1.070	1.097	1.082	1.102	1.085	1.137	1.082	1.103	1.082	1.113	1.066	1.104	1.078	1.107	1.093	1.116	1.075	1.170	1.075	1.168	1.075	1.165	1.080	1.170	1.082	1.168	1.067	1.171
GCA	A	0.555	0.505	0.552	0.490	0.572	0.489	0.540	0.460	0.558	0.487	0.555	0.492	0.548	0.503	0.549	0.496	0.536	0.478	0.463	0.396	0.464	0.394	0.465	0.398	0.465	0.394	0.470	0.394	0.461	0.397
TGT	C	0.358	0.395	0.397	0.347	0.396	0.348	0.370	0.335	0.366	0.349	0.387	0.372	0.392	0.381	0.380	0.371	0.388	0.343	0.365	0.298	0.361	0.298	0.364	0.298	0.366	0.298	0.355	0.298	0.364	0.298
TGC	C	1.642	1.605	1.603	1.653	1.604	1.652	1.630	1.665	1.634	1.651	1.613	1.628	1.608	1.619	1.620	1.629	1.612	1.657	1.635	1.702	1.639	1.702	1.636	1.702	1.634	1.702	1.645	1.702	1.636	1.702
GAT	D	0.261	0.228	0.257	0.229	0.261	0.229	0.260	0.225	0.264	0.229	0.257	0.227	0.274	0.229	0.253	0.228	0.247	0.227	0.262	0.183	0.261	0.182	0.262	0.182	0.263	0.181	0.257	0.181	0.257	0.183
GAC	D	1.739	1.772	1.743	1.771	1.739	1.771	1.740	1.775	1.736	1.771	1.743	1.773	1.726	1.771	1.747	1.772	1.753	1.774	1.738	1.817	1.739	1.818	1.738	1.818	1.737	1.819	1.743	1.819	1.743	1.817
GAG	E	1.596	1.606	1.593	1.609	1.594	1.609	1.601	1.618	1.593	1.609	1.600	1.613	1.599	1.608	1.600	1.613	1.612	1.618	1.623	1.649	1.623	1.650	1.623	1.650	1.623	1.653	1.618	1.645	1.623	1.649
GAA	E	0.404	0.394	0.407	0.391	0.406	0.391	0.399	0.382	0.407	0.391	0.400	0.387	0.401	0.392	0.400	0.387	0.388	0.382	0.377	0.351	0.377	0.350	0.377	0.350	0.377	0.347	0.382	0.355	0.377	0.351
TTT	F	0.450	0.464	0.484	0.435	0.475	0.435	0.468	0.431	0.460	0.435	0.484	0.454	0.477	0.455	0.475	0.458	0.480	0.434	0.378	0.386	0.379	0.386	0.378	0.386	0.380	0.386	0.387	0.384	0.378	0.386
TTC	F	1.550	1.536	1.516	1.565	1.525	1.565	1.532	1.569	1.540	1.565	1.516	1.546	1.523	1.545	1.525	1.542	1.520	1.566	1.622	1.614	1.621	1.614	1.622	1.614	1.620	1.614	1.613	1.616	1.622	1.614
GGT	G	0.299	0.298	0.318	0.284	0.313	0.284	0.300	0.276	0.297	0.284	0.315	0.298	0.316	0.299	0.317	0.296	0.319	0.288	0.302	0.240	0.302	0.241	0.302	0.240	0.302	0.240	0.302	0.243	0.302	0.241
GGG	G	0.930	0.897	0.933	0.892	0.946	0.892	0.929	0.898	0.934	0.892	0.923	0.910	0.917	0.905	0.948	0.909	0.929	0.880	0.850	0.821	0.849	0.817	0.850	0.821	0.850	0.816	0.844	0.820	0.850	0.824
GGC	G	2.212	2.275	2.201	2.295	2.191	2.295	2.219	2.304	2.202	2.295	2.194	2.273	2.201	2.270	2.194	2.274	2.199	2.300	2.301	2.432	2.299	2.433	2.301	2.432	2.301	2.438	2.306	2.430	2.301	2.425
GGA	G	0.559	0.530	0.548	0.529	0.550	0.529	0.552	0.521	0.568	0.529	0.567	0.520	0.566	0.526	0.542	0.521	0.553	0.532	0.548	0.507	0.550	0.509	0.548	0.507	0.548	0.506	0.547	0.507	0.548	0.509
CAC	H	1.697	1.745	1.701	1.746	1.698	1.746	1.682	1.752	1.689	1.746	1.695	1.740	1.705	1.741	1.697	1.743	1.697	1.741	1.752	1.793	1.752	1.793	1.752	1.786	1.752	1.793	1.749	1.793	1.752	1.793
CAT	H	0.303	0.255	0.299	0.254	0.302	0.254	0.318	0.248	0.311	0.254	0.305	0.260	0.295	0.259	0.303	0.257	0.303	0.259	0.248	0.207	0.248	0.207	0.248	0.214	0.248	0.207	0.251	0.207	0.248	0.207
ATT	I	0.534	0.588	0.545	0.581	0.540	0.584	0.534	0.578	0.533	0.581	0.543	0.579	0.542	0.587	0.549	0.578	0.546	0.579	0.350	0.495	0.349	0.495	0.350	0.495	0.350	0.495	0.381	0.495	0.350	0.495
ATA	I	0.138	0.173	0.143	0.177	0.138	0.177	0.138	0.181	0.138	0.177	0.141	0.182	0.138	0.172	0.138	0.169	0.114	0.179	0.138	0.163	0.141	0.163	0.138	0.163	0.138	0.163	0.148	0.163	0.138	0.163
ATC	I	2.328	2.239	2.312	2.243	2.322	2.239	2.328	2.241	2.329	2.243	2.315	2.239	2.320	2.241	2.313	2.253	2.340	2.243	2.512	2.342	2.510	2.342	2.512	2.342	2.512	2.342	2.472	2.342	2.512	2.342
AAA	K	0.345	0.328	0.369	0.304	0.355	0.304	0.333	0.282	0.348	0.304	0.343	0.302	0.340	0.309	0.338	0.303	0.327	0.290	0.260	0.270	0.261	0.265	0.261	0.267	0.261	0.267	0.263	0.267	0.257	0.267
AAG	K	1.655	1.672	1.631	1.696	1.645	1.696	1.667	1.718	1.652	1.696	1.657	1.698	1.660	1.691	1.662	1.697	1.673	1.710	1.740	1.730	1.739	1.735	1.739	1.733	1.739	1.733	1.737	1.733	1.743	1.733
CTA	L	0.295	0.301	0.295	0.293	0.301	0.293	0.288	0.276	0.297	0.293	0.298	0.288	0.306	0.298	0.296	0.296	0.292	0.279	0.231	0.233	0.231	0.233	0.233	0.233	0.232	0.234	0.237	0.233	0.231	0.232
CTC	L	1.167	1.181	1.158	1.185	1.164	1.185	1.162	1.182	1.168	1.185	1.163	1.186	1.155	1.178	1.167	1.184	1.160	1.192	1.196	1.217	1.188	1.216	1.200	1.216	1.188	1.215	1.191	1.218	1.200	1.218
CTG	L	2.171	2.212	2.180	2.225	2.159	2.225	2.180	2.253	2.169	2.225	2.161	2.227	2.168	2.216	2.170	2.221	2.189	2.241	2.272	2.296	2.275	2.297	2.268	2.298	2.279	2.300	2.266	2.295	2.270	2.296
CTT	L	0.367	0.306	0.366	0.296	0.376	0.296	0.370	0.289	0.367	0.296	0.377	0.299	0.371	0.308	0.367	0.299	0.359	0.288	0.300	0.254	0.306	0.254	0.300	0.254	0.301	0.251	0.305	0.254	0.300	0.254
TTA	L	0.262	0.268	0.273	0.270	0.260	0.270	0.258	0.279	0.259	0.270	0.250	0.280	0.267	0.267	0.247	0.275	0.244	0.295	0.259	0.347	0.259	0.347	0.259	0.347	0.259	0.348	0.261	0.347	0.259	0.347
TTG	L	1.738	1.732	1.727	1.730	1.740	1.730	1.742	1.721	1.741	1.730	1.750	1.720	1.733	1.733	1.753	1.725	1.756	1.705	1.741	1.653	1.741	1.653	1.741	1.653	1.741	1.652	1.739	1.653	1.741	1.653
AAC	N	1.762	1.722	1.767	1.714	1.766	1.714	1.757	1.725	1.760	1.714	1.762	1.725	1.750	1.721	1.763	1.727	1.774	1.727	1.791	1.761	1.788	1.761	1.791	1.761	1.791	1.761	1.804	1.761	1.791	1.761
AAT	N	0.238	0.278	0.233	0.286	0.234	0.286	0.243	0.275	0.240	0.286	0.238	0.275	0.250	0.279	0.237	0.273	0.226	0.273	0.209	0.239	0.212	0.239	0.209	0.239	0.209	0.239	0.196	0.239	0.209	0.239
CCA	P	0.353	0.402	0.380	0.387	0.367	0.389	0.356	0.374	0.350	0.390	0.373	0.400	0.353	0.401	0.372	0.402	0.369	0.381	0.312	0.349	0.315	0.347	0.309	0.346	0.312	0.349	0.329	0.348	0.305	0.350
CCC	P	1.426	1.617	1.401	1.655	1.415	1.650	1.426	1.664	1.418	1.638	1.425	1.621	1.419	1.612	1.429	1.622	1.433	1.644	1.498	1.677	1.494	1.687	1.496	1.686	1.492	1.684	1.501	1.684	1.530	1.675
CCT	P	0.535	0.433	0.542	0.430	0.537	0.431	0.550	0.427	0.544	0.433	0.545	0.448	0.536	0.432	0.534	0.437	0.536	0.441	0.510	0.406	0.506	0.404	0.515	0.403	0.498	0.405	0.531	0.404	0.508	0.406
CCG	P	1.686	1.547	1.677	1.528	1.681	1.530	1.668	1.535	1.688	1.539	1.658	1.531	1.692	1.555	1.665	1.540	1.662	1.533	1.679	1.568	1.685	1.561	1.680	1.565	1.697	1.562	1.639	1.563	1.657	1.569
CAA	Q	0.496	0.383	0.513	0.375	0.501	0.379	0.503	0.361	0.497	0.375	0.503	0.381	0.492	0.385	0.505	0.382	0.494	0.359	0.441	0.346	0.447	0.347	0.444	0.345	0.441	0.347	0.448	0.346	0.441	0.345
CAG	Q	1.504	1.617	1.487	1.625	1.499	1.621	1.497	1.639	1.503	1.625	1.497	1.619	1.508	1.615	1.495	1.618	1.506	1.641	1.559	1.654	1.553	1.653	1.556	1.655	1.559	1.653	1.552	1.654	1.559	1.655
AGA	R	0.701	0.751	0.686	0.740	0.711	0.740	0.711	0.671	0.720	0.740	0.711	0.701	0.690	0.751	0.718	0.736	0.734	0.653	0.830	0.681	0.827	0.681	0.834	0.681	0.834	0.685	0.800	0.681	0.794	0.681
AGG	R	1.299	1.249	1.314	1.260	1.289	1.260	1.289	1.329	1.280	1.260	1.289	1.299	1.310	1.249	1.282	1.264	1.266	1.347	1.170	1.319	1.173	1.319	1.166	1.319	1.166	1.315	1.200	1.319	1.206	1.319
CGA	R	0.287	0.236	0.308	0.234	0.294	0.234	0.287	0.222	0.299	0.234	0.292	0.242	0.282	0.240	0.285	0.245	0.278	0.223	0.256	0.233	0.254	0.233	0.253	0.234	0.255	0.230	0.256	0.233	0.271	0.235
CGC	R	2.746	2.925	2.694	2.961	2.711	2.961	2.742	2.999	2.724	2.961	2.716	2.934	2.743	2.918	2.734	2.926	2.759	2.995	2.818	2.956	2.823	2.956	2.818	2.952	2.825	2.959	2.798	2.956	2.805	2.956
CGG	R	0.581	0.505	0.580	0.493	0.591	0.493	0.577	0.484	0.581	0.493	0.578	0.496	0.585	0.517	0.573	0.496	0.574	0.489	0.574	0.521	0.570	0.521	0.577	0.524	0.565	0.522	0.575	0.521	0.574	0.519
CGT	R	0.386	0.333	0.419	0.312	0.403	0.312	0.395	0.296	0.396	0.312	0.413	0.328	0.389	0.325	0.409	0.333	0.389	0.292	0.352	0.290	0.353	0.290	0.352	0.290	0.354	0.289	0.372	0.290	0.350	0.290
AGC	S	1.629	1.707	1.604	1.712	1.625	1.712	1.610	1.726	1.623	1.712	1.617	1.716	1.637	1.708	1.618	1.712	1.631	1.722	1.683	1.754	1.685	1.754	1.683	1.754	1.683	1.754	1.651	1.756	1.683	1.754
AGT	S	0.371	0.293	0.396	0.288	0.375	0.288	0.390	0.274	0.377	0.288	0.383	0.284	0.363	0.292	0.382	0.288	0.369	0.278	0.317	0.246	0.315	0.246	0.317	0.246	0.317	0.246	0.349	0.244	0.317	0.246
TCA	S	0.280	0.167	0.298	0.166	0.277	0.166	0.292	0.167	0.278	0.167	0.296	0.180	0.287	0.173	0.291	0.169	0.258	0.179	0.198	0.180	0.202	0.180	0.198	0.180	0.198	0.180	0.224	0.180	0.195	0.180
TCC	S	1.443	1.530	1.385	1.535	1.440	1.531	1.448	1.537	1.437	1.533	1.439	1.530	1.461	1.517	1.432	1.533	1.471	1.546	1.510	1.555	1.513	1.557	1.506	1.555	1.507	1.556	1.489	1.559	1.530	1.553
TCG	S	1.764	1.742	1.778	1.739	1.757	1.743	1.748	1.757	1.759	1.741	1.762	1.740	1.749	1.745	1.767	1.743	1.750	1.737	1.817	1.780	1.811	1.779	1.821	1.780	1.819	1.778	1.810	1.780	1.800	1.779
TCT	S	0.514	0.561	0.540	0.560	0.526	0.560	0.511	0.538	0.527	0.560	0.504	0.550	0.503	0.565	0.510	0.555	0.521	0.538	0.475	0.485	0.475	0.484	0.475	0.485	0.476	0.485	0.477	0.481	0.474	0.488
ACC	T	1.177	1.146	1.171	1.142	1.181	1.143	1.181	1.152	1.178	1.145	1.173	1.139	1.161	1.136	1.170	1.144	1.216	1.156	1.283	1.159	1.277	1.159	1.290	1.159	1.268	1.159	1.274	1.160	1.280	1.160
ACA	T	0.499	0.539	0.524	0.521	0.511	0.522	0.514	0.511	0.496	0.523	0.534	0.533	0.538	0.528	0.526	0.533	0.486	0.502	0.456	0.394	0.455	0.394	0.449	0.394	0.457	0.394	0.485	0.395	0.478	0.395
ACG	T	1.876	1.927	1.867	1.949	1.866	1.946	1.869	1.955	1.877	1.943	1.858	1.945	1.857	1.938	1.870	1.931	1.879	1.960	1.867	2.044	1.870	2.044	1.872	2.044	1.879	2.044	1.854	2.042	1.855	2.042
ACT	T	0.448	0.388	0.437	0.389	0.442	0.389	0.435	0.383	0.449	0.390	0.435	0.384	0.444	0.398	0.434	0.392	0.419	0.382	0.394	0.403	0.398	0.403	0.389	0.403	0.396	0.403	0.387	0.404	0.386	0.404
GTT	V	0.319	0.270	0.336	0.248	0.337	0.248	0.330	0.234	0.326	0.248	0.343	0.263	0.353	0.275	0.322	0.266	0.318	0.242	0.274	0.245	0.281	0.247	0.274	0.245	0.275	0.245	0.275	0.245	0.272	0.245
GTG	V	2.262	2.386	2.273	2.399	2.254	2.399	2.250	2.414	2.251	2.399	2.263	2.401	2.268	2.390	2.274	2.398	2.275	2.410	2.400	2.498	2.402	2.497	2.397	2.498	2.407	2.499	2.421	2.497	2.406	2.498
GTC	V	1.099	1.036	1.085	1.039	1.094	1.039	1.094	1.048	1.105	1.039	1.082	1.034	1.059	1.027	1.077	1.031	1.096	1.046	1.064	1.036	1.060	1.034	1.064	1.036	1.068	1.035	1.058	1.037	1.058	1.036
GTA	V	0.319	0.308	0.307	0.314	0.315	0.314	0.326	0.304	0.318	0.314	0.313	0.303	0.320	0.308	0.326	0.305	0.312	0.302	0.261	0.221	0.257	0.222	0.265	0.221	0.249	0.221	0.245	0.221	0.264	0.221
TAC	Y	1.784	1.771	1.786	1.771	1.784	1.771	1.777	1.770	1.784	1.771	1.773	1.773	1.774	1.773	1.776	1.773	1.777	1.773	1.772	1.799	1.774	1.799	1.772	1.799	1.772	1.799	1.777	1.799	1.774	1.799
TAT	Y	0.216	0.229	0.214	0.229	0.216	0.229	0.223	0.230	0.216	0.229	0.227	0.227	0.226	0.227	0.224	0.227	0.223	0.227	0.229	0.201	0.226	0.201	0.229	0.201	0.229	0.201	0.223	0.201	0.226	0.201

**Table 5 pathogens-10-01649-t005:** Occurrence of tRNA genes in human host cells for most favored codons for amino acids except Met and Trp in MC viruses.

Amino Acids	Most Favored Codons in MCV	tRNA Isotypes in Human Cells (Khandia et al., 2019; http://gtrnadb.ucsc.edu/Hsapi19/Hsapi19-gene-list.html Accessed on 30 August 2021)
Ala (A)	GCG	AGC (22), GGC (0), CGC (4), UGC (8)
Gly (G)	GGC	ACC (0), GCC (14), CCC (5), UCC (9)
Pro (P)	CCG	AGG (9), GGG (0), CGG (4), UGG (7)
Thr (T)	ACG	AGU (9), GGU (0), CGU (5), UGU (6)
Val (V)	GTG	AAC (9), GAC (0), CAC (11), UAC (5)
Ser (S)	TCG	AGA (9), GGA (0), CGA (4), UGA (4), ACU (0), GCU (8)
Arg (R)	CGC	ACG (7), GCG (0), CCG (4), UCG (6), CCU (5), UCU (6)
Leu (L)	CTG	AAG (9), GAG (0), CAG (9), UAG (3), CAA (6), UAA (4)
Phe (F)	TTC	AAA (0), GAA (10)
Asn (N)	AAC	AUU (0), GUU (20)
Lys (K)	AAG	CUU (15), UUU (12)
Asp (D)	GAC	AUC (0), GUC (13)
Glu (E)	GAG	CUC (8), UUC (7)
His (H)	CAC	AUG (0), GUG (10)
Gln (Q)	CAG	CUG (13), UUG (6)
Ile (I)	ATC	AAU (14), GAU(3), UAU (5)
Tyr (Y)	TAC	AUA (0), GUA (13)
Cys (C)	TGC	ACA (0), GCA (29)

**Table 6 pathogens-10-01649-t006:** Correlation analysis between major axes of variation, silent base contents and significant codon usage indices.

Strains	Axes	A-3	T-3	G-3	C-3	GC3	ENC	CAI	Length
MH320547.1 (MCV 1)	Axis 1	−0.9314 *	−0.9093 *	0.5958 *	0.8538 *	0.9682 *	−0.8997 *	−0.7619 *	0.2394 *
	Axis 2	−0.1276	−0.1968 *	−0.0768	0.2652 *	0.1566	−0.1527	−0.2341 *	−0.0137
	Axis 3	0.2276 *	0.1862 *	−0.1975 *	−0.1666 *	−0.2225 *	0.2063 *	0.1416	0.0672
	Axis 4	0.1543	0.1586 *	0.0408	−0.2251 *	−0.1753 *	0.1385	0.1452	−0.0662
	Axis 5	0.1707 *	0.2419 *	−0.2423 *	−0.1012	−0.2181 *	0.2414 *	0.0552	0.0088
	Axis 6	0.0092	0.0419	−0.0401	−0.0358	−0.0239	−0.0288	−0.0372	0.0576
	Axis 7	0.0388	0.1246	−0.2989 *	0.0667	−0.0828	0.1339	0.0133	−0.0680
MH320552.1 (MCV 1)	Axis 1	−0.9296 *	−0.9131 *	0.5715 *	0.8528 *	0.9685 *	−0.8969 *	−0.7561 *	0.2290 *
	Axis 2	−0.0582	−0.1120	−0.2011 *	0.2259 *	0.0773	−0.0440	−0.1913 *	−0.0248
	Axis 3	0.2503 *	0.2010 *	−0.2073 *	−0.1907 *	−0.2427 *	0.2229 *	0.1696 *	0.0472
	Axis 4	0.1702 *	0.1900 *	−0.0512	−0.1833 *	−0.2019 *	0.1852 *	0.0919	−0.0865
	Axis 5	−0.0481	−0.0445	0.0607	0.0271	0.0563	−0.1229	−0.0647	0.0993
	Axis 6	−0.0760	−0.0954	0.0129	0.1074	0.0881	−0.0526	0.0225	−0.0956
	Axis 7	0.0645	0.1880 *	−0.3423 *	0.0410	−0.1214	0.1816 *	0.0113	−0.0512
MH320553.1 (MCV 1)	Axis 1	−0.9314 *	−0.9092 *	0.5860 *	0.8530 *	0.9684 *	−0.8987 *	−0.7569 *	0.2377 *
	Axis 2	−0.1374	−0.2091 *	−0.0810	0.2798 *	0.1681 *	−0.1635 *	−0.2513 *	−0.0121
	Axis 3	0.2229 *	0.1794 *	−0.1786 *	−0.1648 *	−0.2231 *	0.1883 *	0.1021	0.0820
	Axis 4	0.0473	0.0345	−0.1852 *	0.0531	−0.0338	0.0594	0.0062	0.0163
	Axis 5	0.1509	0.1896 *	−0.2074 *	−0.0759	−0.1868 *	0.2458 *	0.0652	−0.0697
	Axis 6	0.0448	0.1114	−0.0344	−0.0810	−0.0795	0.0565	−0.0434	0.0625
	Axis 7	0.0228	0.1476	−0.2944 *	0.0644	−0.0800	0.1398	0.0087	−0.0453
MH320554.1 (MCV 1)	Axis 1	−0.9285 *	−0.9185 *	0.5821 *	0.8487 *	0.9683 *	−0.8994 *	−0.7633 *	0.2513 *
	Axis 2	−0.2313 *	−0.2993 *	0.1430	0.2486 *	0.2722 *	−0.2667 *	−0.2705 *	0.0119
	Axis 3	0.1055	0.0996	−0.3217 *	0.0477	−0.1125	0.1453	−0.0523	0.0022
	Axis 4	−0.1650 *	−0.2137 *	−0.0513	0.2648 *	0.2126 *	−0.1830 *	−0.1922 *	0.0952
	Axis 5	0.0285	0.0882	−0.1460	0.0326	−0.0673	0.1308	−0.0189	−0.0775
	Axis 6	0.0493	0.1502	−0.0524	−0.1025	−0.0990	0.0765	0.0059	0.0302
	Axis 7	−0.0235	0.0783	−0.3056 *	0.1282	−0.0291	0.0843	−0.0048	−0.0619
MH320555.1 (MCV 1)	Axis 1	−0.9320 *	−0.9107 *	0.5884 *	0.8516 *	0.9689 *	−0.9011 *	−0.7589 *	0.2581 *
	Axis 2	0.1935 *	0.2929 *	−0.0121	−0.3093 *	−0.2397 *	0.2445 *	0.3075 *	−0.0083
	Axis 3	0.1567	0.1426	−0.2940 *	−0.0207	−0.1627 *	0.1764 *	0.0133	0.0084
	Axis 4	0.1847 *	0.1993 *	−0.0439	−0.2218 *	−0.2120 *	0.1867 *	0.1742 *	−0.0830
	Axis 5	0.0811	0.1479	−0.1421	−0.0446	−0.1286	0.1767 *	0.0234	−0.0533
	Axis 6	0.0340	0.1023	−0.0170	−0.0832	−0.0697	0.0371	−0.0294	0.0307
	Axis 7	−0.0222	0.1013	−0.2641 *	0.0971	−0.0341	0.0998	−0.0015	−0.0451
KY040275.1 (MCV 1)	Axis 1	−0.9310 *	−0.9133 *	0.5876 *	0.8513 *	0.9698 *	−0.8994 *	−0.7607 *	0.2169 *
	Axis 2	0.0317	0.0060	−0.2354 *	0.1171	−0.0303	0.0457	−0.1273	0.0209
	Axis 3	−0.1911 *	−0.2807 *	0.0609	0.2568 *	0.2453 *	−0.2270 *	−0.2383 *	−0.0807
	Axis 4	0.0758	−0.0583	−0.1103	0.0286	−0.0067	0.0096	0.0535	0.0363
	Axis 5	−0.0768	−0.0901	0.1390	0.0204	0.0921	−0.1580	−0.0281	0.0996
	Axis 6	0.0462	0.0847	−0.0199	−0.0918	−0.0694	0.0478	−0.0289	0.0245
	Axis 7	0.0103	0.1729 *	−0.3222 *	0.0810	−0.0794	0.1612 *	0.0315	−0.1102
KY040276.1 (MCV 1)	Axis 1	−0.9345 *	−0.9155 *	0.5907 *	0.8551 *	0.9693 *	−0.9001 *	−0.7566 *	0.2211 *
	Axis 2	−0.0008	−0.0363	−0.1378	0.1180	0.0113	−0.0220	−0.1447	0.0127
	Axis 3	0.2975 *	0.2979 *	−0.1643 *	−0.3068 *	−0.3122 *	0.2993 *	0.2681 *	0.0147
	Axis 4	−0.0164	−0.0577	−0.1468	0.1270	0.0460	0.0045	−0.0663	0.0127
	Axis 5	0.1076	0.1202	−0.1726 *	−0.0239	−0.1286	0.1864 *	−0.0352	−0.0871
	Axis 6	0.0328	0.0730	−0.0057	−0.0718	−0.0568	0.0310	−0.0431	0.0414
	Axis 7	0.0411	0.1270	−0.3367 *	0.0731	−0.0925	0.1367	−0.0089	−0.0656
KY040277.1 (MCV 1)	Axis 1	−0.9324 *	−0.9153 *	0.5772 *	0.8514 *	0.9679 *	−0.8985 *	−0.7705 *	0.1989 *
	Axis 2	0.2372 *	0.3087 *	−0.0301	−0.3321 *	−0.2764 *	0.2624 *	0.3060 *	−0.0167
	Axis 3	0.2291 *	0.2053 *	−0.2563 *	−0.1183	−0.2355 *	0.2432 *	0.0676	0.0602
	Axis 4	−0.0482	−0.0728	−0.1454	0.1664 *	0.0724	−0.0210	−0.0980	0.0464
	Axis 5	0.0649	0.1125	−0.1634 *	0.0045	−0.1047	0.1831 *	−0.0211	−0.1143
	Axis 6	0.0512	0.1469	−0.0823	−0.0757	−0.0979	0.1033	−0.0157	−6.94 × 10^−5^
	Axis 7	0.0142	0.1340	−0.2789 *	0.0603	−0.0793	0.1333	0.0073	−0.1334
U60315.1 (MCV 1)	Axis 1	−0.9261 *	−0.9181 *	0.5844 *	0.8321 *	0.9643 *	−0.8935 *	−0.7533 *	0.1458
	Axis 2	−0.0746	−0.0615	−0.2056 *	0.2064 *	0.0685	−0.0559	−0.1630 *	0.1041
	Axis 3	−0.2048 *	−0.2285 *	−0.0010	0.2678 *	0.2313 *	−0.1733 *	−0.2219 *	−0.0541
	Axis 4	−0.0054	−0.1029	−0.0276	0.0963	0.0609	−0.0562	0.0090	0.0370
	Axis 5	0.1431	0.2410 *	−0.3398 *	−0.0012	−0.2028 *	0.2655 *	0.1321	−0.0781
	Axis 6	0.0828	0.1614	−0.2440 *	−0.0049	−0.1209	0.1391	−0.0608	−0.0230
	Axis 7	0.1146	0.0702	0.1049	−0.1428	−0.1003	0.0450	0.0366	0.2033 *
MH320548.1 (MCV 2)	Axis 1	−0.9255 *	−0.9464 *	0.6122 *	0.8414 *	0.9765 *	−0.9038 *	−0.8090 *	0.2244 *
	Axis 2	0.1193	0.2168 *	−0.2818 *	−0.0169	−0.1809 *	0.1442	0.0864	0.0660
	Axis 3	0.1119	0.1541	−0.1354	−0.0890	−0.1588	0.1422	0.0865	−0.0322
	Axis 4	−0.0111	0.0530	0.0861	−0.0368	−0.0211	0.0270	0.0267	−0.1704 *
	Axis 5	−0.1085	0.0357	−0.1608 *	0.1043	0.0357	−0.0009	−0.2263 *	0.0848
	Axis 6	−0.0136	−0.0315	0.1342	−0.0709	0.0246	−0.0517	0.1175	−5.83 × 10^−5^
	Axis 7	0.2027 *	−0.0331	0.1212	−0.1616 *	−0.0892	−0.0472	0.0115	0.1864 *
MH320549.1 (MCV 2)	Axis 1	−0.9254 *	−0.9457 *	0.6181 *	0.8417 *	0.9773 *	−0.9064 *	−0.8064 *	0.2252 *
	Axis 2	0.0780	0.1758 *	−0.2696 *	0.0225	−0.1410	0.1143	0.0502	0.0888
	Axis 3	0.1014	0.1374	−0.0956	−0.0980	−0.1449	0.1293	0.0803	−0.0154
	Axis 4	−0.0275	0.0441	0.0868	−0.0156	−0.0069	0.0161	0.0191	−0.1741 *
	Axis 5	−0.1107	0.0429	−0.2004 *	0.1168	0.0298	0.0085	−0.2278 *	0.0885
	Axis 6	−0.0753	−0.0278	0.0578	9.65 × 10^−5^	0.0508	−0.0420	0.0967	−0.0216
	Axis 7	0.1652 *	−0.0669	0.1527	−0.1395	−0.0514	−0.0840	0.0057	0.1888 *
MH320550.1 (MCV 2)	Axis 1	−0.9262 *	−0.9463 *	0.6108 *	0.8417 *	0.9766 *	−0.9037 *	−0.8086 *	0.2227 *
	Axis 2	0.1232	0.2178 *	−0.2932 *	−0.0124	−0.1837 *	0.1440	0.0917	0.0689
	Axis 3	0.1158	0.1589	−0.1283	−0.0959	−0.1629 *	0.1449	0.0844	−0.0283
	Axis 4	−0.0147	0.0558	0.07813	−0.0346	−0.0210	0.0279	0.0201	−0.1706 *
	Axis 5	−0.1078	0.0340	−0.1625 *	0.1090	0.0385	−0.0034	−0.2270 *	0.0940
	Axis 6	−0.0098	−0.0249	0.1383	−0.0755	0.0208	−0.0455	0.1331	−0.0101
	Axis 7	0.1915 *	−0.0446	0.1326	−0.1551	−0.0781	−0.0600	0.0168	0.1912 *
MH320551.1 (MCV 2)	Axis 1	−0.9253 *	−0.9469 *	0.6113 *	0.8426 *	0.9772 *	−0.9075 *	−0.8111 *	0.2307 *
	Axis 2	0.1145	0.2244 *	−0.2886 *	−0.0218	−0.1832 *	0.1572	0.0894	0.0562
	Axis 3	0.1262	0.1555	−0.1430	−0.0939	−0.1691 *	0.1547	0.1029	−0.0444
	Axis 4	−0.0002	0.0742	0.0669	−0.0450	−0.0382	0.0401	0.0234	−0.1674 *
	Axis 5	−0.1008	0.0384	−0.1703 *	0.0989	0.0275	0.0132	−0.2299 *	0.0811
	Axis 6	0.0429	−0.0549	0.1081	−0.0839	0.0027	−0.0693	0.0759	0.0733
	Axis 7	0.2219 *	0.0099	0.0299	−0.1462	−0.1171	0.0020	−0.0063	0.1292
MH320556.1 (MCV 2)	Axis 1	−0.9261 *	−0.9449 *	0.6129 *	0.8431 *	0.9770 *	−0.9057 *	−0.8084 *	0.2248 *
	Axis 2	0.1206	0.2180 *	−0.2917 *	−0.0128	−0.1797 *	0.1452	0.0832	0.0690
	Axis 3	0.1146	0.1617 *	−0.1333	−0.0925	−0.1612 *	0.1475	0.0849	−0.0366
	Axis 4	−0.0062	0.05624	0.0752	−0.0377	−0.0275	0.0334	0.0343	−0.1755 *
	Axis 5	−0.1016	0.0416	−0.1704 *	0.1021	0.0303	0.0087	−0.2214 *	0.0875
	Axis 6	−0.0106	−0.0361	0.1142	−0.0661	0.0228	−0.0469	0.1061	−0.0031
	Axis 7	0.2275 *	−0.0160	0.0886	−0.1673 *	−0.1144	−0.0287	0.0205	0.1858 *
KY040274.1 (MCV 2)	Axis 1	−0.9256 *	−0.9458 *	0.6106 *	0.8409 *	0.9766 *	−0.9020 *	−0.8044 *	0.2229 *
	Axis 2	0.1273	0.2160 *	−0.2898 *	−0.0149	−0.1848 *	0.1428	0.0814	0.0686
	Axis 3	0.1108	0.1575	−0.1311	−0.0933	−0.1606 *	0.1428	0.0868	−0.0289
	Axis 4	−0.0124	0.0560	0.0755	−0.0324	−0.0228	0.0278	0.0192	−0.1690
	Axis 5	−0.0870	0.0306	−0.1563	0.0920	0.0285	−0.0047	−0.2359 *	0.1069
	Axis 6	−0.0298	−0.0158	0.1230	−0.0591	0.0272	−0.0349	0.1317	−0.0227
	Axis 7	0.1767 *	−0.0647	0.1402	−0.1392	−0.0601	−0.0821	0.0191	0.1894 *

* level of significance was measured at *p* < 0.05.

**Table 7 pathogens-10-01649-t007:** Details of selected strains of MCV for the present study.

Subtype	Accession ID	Country of Isolation	Total Number of CDS	Selected CDS	Genome Size
MCV 1	MH320547.1	Slovenia	178	148	187,826 bp
MCV 1	MH320552.1	Slovenia	176	147	187,884 bp
MCV 1	MH320553.1	Slovenia	178	148	187,558 bp
MCV 1	MH320554.1	Slovenia	177	147	196,781 bp
MCV 1	MH320555.1	Slovenia	177	148	189,292 bp
MCV 1	KY040275.1	Spain	181	144	188,253 bp
MCV 1	KY040276.1	Spain	179	148	189,098 bp
MCV 1	KY040277.1	Spain	179	146	188,458 bp
MCV 1	U60315.1	Not specified	163	140	190,289 bp
MCV 2	MH320548.1	Slovenia	170	144	190,319 bp
MCV 2	MH320549.1	Slovenia	170	144	193,271 bp
MCV 2	MH320550.1	Slovenia	170	144	196,206 bp
MCV 2	MH320551.1	Slovenia	170	144	192,156 bp
MCV 2	MH320556.1	Slovenia	170	144	189,257 bp
MCV 2	KY040274.1	Spain	170	144	192,183 bp

## Data Availability

Not applicable.

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
