# Peer review of "Strategies and Patterns of Codon Bias in Molluscum Contagiosum Virus"

_pathogens, 2021, doi:10.3390/pathogens10121649_

Round 1

Reviewer 1 Report

Molluscum contagiosum viruses belong to the Poxviridae family, and cause self-limiting skin disease in humans. Their genome, while sharing genes with them, lacks genes widely present in other poxviruses and presents other genes that are unique in the poxviridae family. These characteristic features may well provide them a way to co-exist with their host.

The authors have analysed MCV-1 and MCV-2 genomes in order to determine the forces behind codon usage in Molluscum contagiosum virus.

This article gives an interesting insight into mechanisms governing viruses CDS evolution.

Minor changes are required:

- Lines 188-190: “No strand specific bias was observed in synonymous codon usage (Table 4). MCV-1 and MCV-2 genomes exhibited preference towards G/C ending rather than A/T ending codons in coding amino acids except methionine and tryptophan.”

This sentence is not clear, since both Met and Trp exhibit G ending. If the authors mean that these two amino acids are not concerned by bias because they are only coded by one codon, it might be clearer to say so.

- Table 3: M appears (with a value of 1.000 for ATG), but not W. Why include one and not the other (they both are present in table 4)?

Table 5: Regarding tRNA isotypes, some sequences with T instead of U have been left at the end of the table

- Line 43-44: “as reported in Poxviridae family [6, 70]”. Ref 70 refers to herpesviruses, not poxviruses.

- Line 70: “MCV infection is slow progressing, self-limiting and generally does not affect […]” (does is missing)

Author Response

We thank the reviewer for critical evaluation and comments. The response for the reviewer's comments are provided below.

  1. - Lines 188-190: “No strand specific bias was observed in synonymous codon usage (Table 4). MCV-1 and MCV-2 genomes exhibited preference towards G/C ending rather than A/T ending codons in coding amino acids except methionine and tryptophan.”

Thanks for the comments. The sentence is made clearer as follows

“MCV-1 and MCV-2 genomes exhibited preference towards G/C ending rather than A/T ending codons in coding amino acids except methionine (Met) and tryptophan (Trp) as Met and Trp are coded by single codons”

  1. - Table 3: M appears (with a value of 1.000 for ATG), but not W. Why include one and not the other (they both are present in table 4)?

From both tables (table 3 and 4), rows corresponding to M and W are removed these amino acids are not connected with codon bias

  1. -Table 5: Regarding tRNA isotypes, some sequences with T instead of U have been left at the end of the table

As suggested, two codons of Ile (AAU, GAU) and that of Tyr (AUA, GUA) are modified.

  1. - Line 43-44: “as reported in Poxviridae family [6, 70]”. Ref 70 refers to herpesviruses, not poxviruses.

The sentence is modified as follows

Significant differences in intragenomic ENC (SD ≥ 5.7) and GC3 (SD ≥ 7.2) and strong positive correlation between ENC and GC3 point out the role of base compositional constraints in shaping SCUB as reported in large double-stranded DNA viruses [6,70].

  1. - Line 70: “MCV infection is slow progressing, self-limiting and generally does not affect […]” (does is missing)

The sentence is modified as follows

MCV infection is slow progressing, self-limiting and generally does not affect immune-competent individuals because immune-competent host cells possess certain immune pathways that can attenuate viruses having high CG (CpG) dinucleotide frequencies [76].

Reviewer 2 Report

This manuscript describes and analyzes codon usage patterns in 3 molluscum contagiosum viral strains. It is very technical and replete with abbreviations.

I have a few questions and points for the authors to consider.

  1. Primary is that the authors should decide whether this is meant for general audience or for experts in the field of protein evolution. If the former, they need to better explain the methods and the nomenclature and abbreviations used. Examples: What is ENC on line 139 and PR2 bias (line 174)? Better simpler explanations of the methods would also be helpful.
  2. Lines 106-7. Do MCV’s (cytoplasmic replicon) become integrated into host genome? Citation?

Author Response

We greatly appreciate reviewer's critical evaluation and comments. Below are the responses for reviewer's comments.

  1. Primary is that the authors should decide whether this is meant for general audience or for experts in the field of protein evolution. If the former, they need to better explain the methods and the nomenclature and abbreviations used. Examples: What is ENC on line 139 and PR2 bias (line 174)? Better simpler explanations of the methods would also be helpful.

I would like to mention that all such abbreviations are expanded in the Materials and methods section part under the corresponding sections.

  1. Lines 106-7. Do MCV’s (cytoplasmic replicon) become integrated into host genome? Citation?

MCV does not integrate into host genome. The sentence is modified as follows

As MCV uses humans as their natural host, long term association with human cells may provide MCVs a platform for their own evolution [58]